# Properties and Composition of Magnetized Nuclei

**V.N. Kondratyev** [1,2]

1    Bogoliubov Laboratory of Theoretical Physics, Joint Institute for Nuclear Research, 141980 Dubna, Russia; vkondrat@theor.jinr.ru
2    Physics Deartment, Dubna State University, Universitetskaya str. 19, 141982 Dubna, Russia

**Abstract:** The properties and mass distribution of the ultramagnetized atomic nuclei which arise in heavy-ion collisions and magnetar crusts, during Type II supernova explosions and neutron star mergers are analyzed. For the magnetic field strength range of 0.1–10 *teratesla*, the Zeeman effect leads to a linear nuclear magnetic response that can be described in terms of magnetic susceptibility. Binding energies increase for open shell and decrease for closed shell nuclei. A noticeable enhancement in theyield of corresponding explosive nucleosynthesis products with antimagic numbers is predicted for iron group and r-process nuclei. Magnetic enrichment in a sampleof $^{44}$Ti corroborate theobservational results and imply a significant increase in the quantity of the main titanium isotope, $^{48}$Ti, in the chemical composition of galaxies. The enhancement of small mass number nuclides in the r-process peak may be due to magnetic effects.

**Keywords:** nucleosynthesis; supernova; magnetars

## 1. Introduction

Radioactive nuclides synthesized during nuclear processes make it possible to probe active regions of nuclear reactions in respective sites, cf., e.g., [1–8]. For example, the radioactive decay of iron group isotopes ($^{44}$Ti, $^{56}$Co, $^{57}$Co) isthe most plausible source of energy [5], which feeds infrared, optical, and ultraviolet radiation insupernova (SN) remnants. The contribution of$^{44}$Ti dominates for SNe older than three or four years, until an interaction of ejecta with the surrounding matter increases and becomesthe dominant source. Accordingly, the light curves and spectra of infrared and ultraviolet radiation were analyzed using complex and model-dependent computer simulations [5]. An estimate ofthe initial mass of $^{44}$Ti in SNR 1987A was made, i.e.,$(1-2)\cdot 10^{-4}\cdot M_{\text{solar}}$ (in solar masses). This value significantly exceeds model predictions (see [9] and below). Neutron star mergers are another plausible source [10,11] of nucleosynthetic components of r-process nuclide enrichment in galactic chemical evolution.

Radioisotopes synthesized in SN explosions can be observed directly by recording the characteristic gamma lines accompanying their decay [1,9]. The radioactive decay chain $^{44}$Ti → $^{44}$Sc → $^{44}$Ca leads to the emission of lines with energies of 67.9 keV and 78.4 keV (from $^{44}$Sc) and 1157 keV (from $^{44}$Ca) of approximately the same intensity. The half-life of $^{44}$Ti, i.e., about 60 years under Earth conditions, allows us to estimate the mass of this isotope in the remnant. The obtained observational values for the total mass of $^{44}$Ti nuclides synthesized in SN explosions significantly exceed model predictions, showing a mass of initially synthesized nuclides of $^{44}$Ti at $M_{\text{Ti}} \sim 10^{-5}\cdot M_{\text{solar}}$ in the absence of magnetic effects. These predictions are consistent with observational data of SN-type I, see [6] and refs. therein. Consideration of specific SN explosion scenarios leads, in some cases, c.f., e.g., [12,13], to mass values approaching those in the observational data.

Superstrong magnetic fields exceeding *teratesla* (TT, 1 TT = $10^{16}$ G) arise in SN explosions [1,2], neutron star mergers [3], heavy ion collisions [4], and magnetar crusts [7], in conjunction with

observations of soft-gamma repeaters and abnormal X-ray pulsars. The nuclides formed in such processes contain information about the structure of matter and the mechanisms of explosive processes. In this contribution, we analyze an effect of a relatively weak magnetic field on nuclear structure, and discuss the possibility of using radionuclides to probe the internal regions of these processes. The next section briefly describes the used methods of nuclear statistical equilibrium for the description and analysis of nucleosynthesis. Section 3 considers changes in the structure and properties of atomic nuclei due to Zeeman splitting of energy levels of nucleons. It is shown that such a mechanism dominates with a magnetic field strength range of 0.1–10 TT, and results in a linear nuclear magnetic response which is in agreement with calculations made using covariant density functional theory, cf. [14,15]. Magnetic susceptibility is a key quantity for the description and analysis of nuclear magnetization. The influence of magnetic fields on the composition of nuclei is considered in Sections 3.2 and 3.3. Conclusions are presented in Section 4.

## 2. Abundance of Atomic Nuclei at Statistical Equilibrium

Nuclear statistical equilibrium (NSE) approximation has been used very successfully to describe the abundance of iron group nuclei and nearby nuclides for more than half a century. Under NSE conditions, the yield of nuclides is mainly determined by the binding energy of the resulting atomic nuclei. The magnetic effects in NSE were considered in [1,2,8]. Recall that at temperatures ($T \leq 10^{9.5}$ K) and field strengths ($H \geq 0.1$ TT), the dependence on the magnetic field of the relative yield $y = Y(H)/Y(0)$ is determined mainly by a change of nuclear binding energy, $\Delta B$, in a field, and can be written in the following form:

$$y = \exp\{\Delta B/kT\}, \tag{1}$$

The binding energy of a nucleus is given in the form of the energy difference between noninteracting free nucleons $E_N$ and the nucleus consisting of them, i.e., $E_A$, $B = E_N - E_A$. Under conditions of thermodynamic equilibrium at temperature $T$, the corresponding energy is expressed as follows:

$$E = \frac{kT^2}{\Sigma} \frac{\partial \Sigma}{\partial T} \tag{2}$$

in terms of a partition function $\Sigma = \sum_i \exp\{-e_i/kT\}$, where $e_i$ denotes the energy of nuclear particles in an $i$-state and $k$ is the Boltzmann constant. Using Equation (2) for free nucleons, the energy component due to an interaction with a magnetic field can be written in the following form: $E_\alpha = -\frac{g_\alpha}{2}\omega_L\text{th}(g_\alpha\omega_L/2kT)$, where th$(x)$ is the hyperbolic tangent and the Larmor frequency $\omega_L = \mu_N H$. Here, the well-known [16] spin $g-$ factors $g_p \approx 5.586$ and $g_n \approx -3.826$ for protons $\alpha = $ p and neutrons $\alpha = $ n. For values of temperature ($T \sim 10^{9.5}$ K) and field strengths ($H \sim 1$ TT), here, one gets $E_\alpha \sim -10^{0.5}$ keV.

## 3. Synthesis of Ultramagnetized Atomic Nuclei

The Zeeman—Paschen—Back effect is associated with a shift of nucleon energy levels due to an interaction of nucleon magnetic moments with a field. Dramatic change in nuclear structure occurs under conditions of nuclear level crossing [2,8]. The characteristic energy interval $\Delta\varepsilon \sim 1$ MeV determines the scale of a field strength, i.e., $\Delta H_{\text{cross}} \sim \Delta\varepsilon/\mu_N \sim 10^{1.5}$ TT, at which nonlinear effects dominate. Here, $\mu_N$ stands for nuclear magneton. In case of a small field strength, i.e., $H \ll 10^{1.5}$ TT, a linear approximation can be used. At field intensities $H \geq 0.1$TT, one can neglect the residual interaction [8]. Under such conditions, the total value of a nucleon spin quantum number on a subshell (and a nucleus) is the maximum possible, similar to the Hund rule, which is well known for the electrons of atoms.

### 3.1. Zeeman Energy in Atomic Nuclei

The self-consistent mean field is a widely used approach for obtaining realistic descriptions and analyses of the properties of atomic nuclei. The single-particle (sp) Hamiltonian $\hat{H}_\alpha$ for nuclei in a relatively weak magnetic field $H$ within the linear approximation can be written as

$$H_\alpha = H_\alpha{}^0 - \left(g_\alpha{}^o\hat{l} + g_\alpha\hat{s}\right)\omega_L \tag{3}$$

for protons $\alpha = p$ and neutrons $\alpha = n$. Here, $\hat{H}_\alpha^0$ represents the sp Hamiltonian for isolated nuclei, while the orbital moment and spin operators are denoted by $\hat{l}$ and $\hat{s}$, respectively. The interaction of dipole nucleon magnetic moments with a field is represented by terms containing the vector $\omega_L = \mu_N H$, and $g_\alpha^o$ denotes orbital $g-$ factors $g_p^o = 1$ and $g_n^o = 0$.

Thus, the binding energy decreases for magic nuclei with a closed shell, zero magnetic moment and, therefore, zero interaction energy with a magnetic field. In cases of antimagic nuclei with open shells, a significant (maximum possible under these conditions) magnetic moment leads to an additional increase in the binding energy $B$ in a field. In this case, the leading component of such a magnetic contribution is represented by the sum over the filled $i$ sp energy levels $\varepsilon_i$, $B_m = \sum_{i-occ} \varepsilon_i$, see [8]. In the representation of angular momentum for spherical nuclei, the sp states $|i>$ are conveniently characterized by quantum numbers (see [16]): $n$-radial quantum number, angular momentum $l$, total spin $j$, and spin projection on the direction of the magnetic field $m_j$. Then, using sp energies $\varepsilon_{nljm_j}$ and wave functions $|nljm_j>$, the magnetic energy change $\Delta B^m = B^m(H) - B^m(0)$ in a field $H$ can be written as

$$\Delta B_\alpha^m = \kappa_\alpha \omega_L, \kappa_\alpha = \sum_{i-occ} \kappa_\alpha^i,$$
$$\kappa_\alpha^i = \sum_{m,s} | < lm, \tfrac{1}{2}s|jm_j > |^2 (g_\alpha^o m + g_\alpha s)$$
$$= \begin{cases} (g_\alpha^o l + g_\alpha/2)m_j/j, & \text{for } j = l + 1/2, \\ (g_\alpha^o(l+1) - g_\alpha/2)m_j/(j+1), & \text{for } j = l - 1/2, \end{cases} \tag{4}$$

where $(\alpha = n, p)$, $< lm, \tfrac{1}{2}s|jm_j >$ is the Clebsh-Gordan coefficient. The result from Equation (4) is similar to that obtained in the Schmidt model [16]. We stress here that in this case, parameter $\kappa_\alpha$ is given by the combined susceptibility of all the independent nucleons spatially confined due to a mean field. The linear response regime at magnetic induction $H < 10$ TT is also confirmed by consideration within the covariant density functional theory, cf. [14,15]. The present analysis in terms of magnetic susceptibility yields transparent and clear results for nuclear magnetic reactivity with fundamental consequences for the study of nuclear structure and dynamics in strong magnetic fields.

### 3.2. Iron Region

In a case of magic numbers, $\kappa = 0$ (see Figure 1a). The dependence on the magnetic field in the synthesis of nuclei is due to a change in the energy of an interaction of free nucleons with a field. The magnetization of a nondegenerate nucleon gas and the arising component of magnetic pressure lead to an effective decrease in the binding energy of magic nuclei and, as a result, to the suppression of the yield of corresponding chemical elements. However, we notice that the suppression factor is less significant in the case of realistic magnetic field geometry, see [2]. A significant magnetic moment and parameter $\kappa$ contribute to an increase in binding of nucleons for ultramagnetized antimagic nuclei in a field. The increase in nucleosynthesis products caused by such an enhancement is weakly sensitive to the structure of a magnetic field [2].

Let us consider the normalized yield coefficient of antimagic even–even symmetric nuclei of the $1f_{7/2}$ and $2p_{3/2}$ shells and the double magic nucleus $^{56}$Ni, i.e., $[i/\text{Ni}] \equiv y_i/y_{\text{Ni}}$. As is seen in Figure 2, the volume of synthesis of $^{44}$Ti and $^{48}$Cr increases sharply with increasing magnetic induction, whereas the output of $^{52}$Fe varies relatively insignificantly, and the total mass of $^{60}$Zn is almost constant. It is important to recall the mysteriously large abundance of titanium obtained in direct observations of

SN-type II remnants; see refs. [2,5,9]. Observational data suggest a $^{44}$Ti nucleus yield for type II SNe far exceeding model predictions and similar results for type I SNe. As one can see from Equations (3) and (4) and Figure 1b, the magnetic increase in the synthesis of nuclides by an order of magnitude corresponds to a field strength of several TT. Such magnetic induction is consistent with simulation predictions and an explosion energy of SNe [1,2].

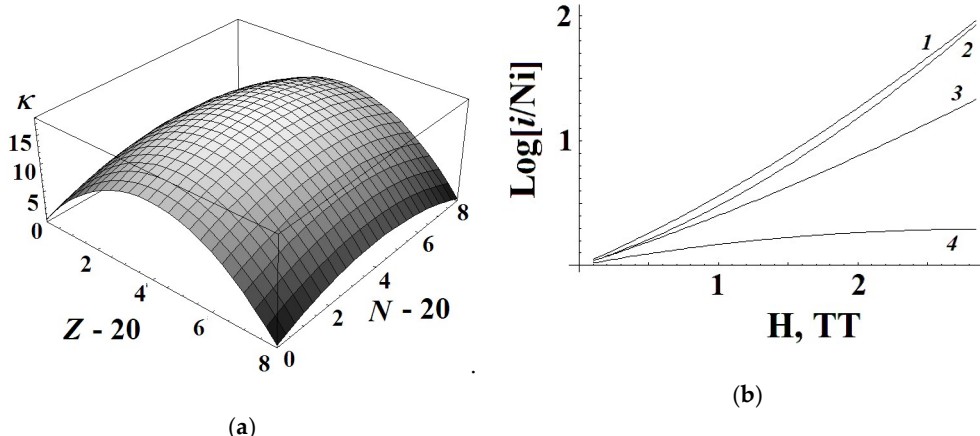

(a)　　　　　　　　　　　　　　　　　(b)

**Figure 1.** Magnetic effects for nuclei in the iron region: (**a**) Dependence on the number of protons and neutrons of the magnetic susceptibility for nuclei with filled $1f_{7/2}$ shell. The minimum values $\kappa_{\text{magic}} = 0$ correspond to double magic nuclei at $Z(\&N) = 20$ or 28, the maximum value $\kappa_{\text{max}} \approx 17.51$ for the antimagic nucleus $^{48}$Cr at $Z = N = 24$; (**b**) Magnetic field dependence of the yield ratio $[i/\text{Ni}]$ (see text) for $^{56}$Ni and $^{44}$Ti—2, $^{48}$Cr—1, $^{52}$Fe—3, $^{60}$Zn—4, at $kT$=0.5 MeV.

Accounting for Equations (1) and (4) and Figure 1b, we notice that such conditions suggest even stronger enrichment of $^{48}$Cr, since the maximum magnetic susceptibility corresponds to a half-filled shell. In the case of the filling of shell $1f_{7/2}$ (iron group nuclei), this condition is satisfied at $Z = N = 24$ (see Section 3.1). Then, a significant value of parameter $\kappa_{\text{Cr}} = 17.51$ leads to a noticeable magnetic amplification of the synthesis of $^{48}$Cr nuclide. The chain of radioactive decay $^{48}$Cr $\to$ $^{48}$V $\to$ $^{48}$Ti generates an excess of the dominant titanium isotope.

*3.3. The r-Process Nuclides*

r-process nuclides can plausibly originate from neutron star mergers. In a single event, such sites produce 100 times larger nuclide volumes than Type II SN events. In the first stage of the production of r-process nuclei, matter undergoes explosive burning and is heated to conditions of NSE equilibrium [11]; the abundance is given by Equation (1). Significantly amplified magnetic induction can affect nucleosynthesis processes in both cases. As is seen in Equation (4), a noticeable magnetic modification in nuclear properties is expected for mass numbers corresponding to pronounced magic numbers, i.e., N&Z = 50, 82, and 126.

As is illustrated in Figure 2a, for mass numbers $A$ = 40—100, considerable values of magnetic susceptibility are displayed for nuclei corresponding to $1f_{7/2}$ and $1g_{9/2}$ shells. Neutron number $N = 50$ gives a magic number for the concentration of nuclear material, as with r-process scenarios. Such a mass enhancement also originates from a small cross section of ($n$, $\gamma$) reactions on magic nuclei, see [17]. The normalized yield coefficients of some nuclei of the $1g_{9/2}$ shell and the double magic nucleus $^{100}$Sn, i.e., $[i/\text{Sn}] = y_i/y_{\text{Sn}}$ are presented in Figure 2b. As is shown in Figure 2b, the magnetic effects give rise to an enrichment of nuclear components with smaller mass numbers. However, $N = 50$ isotone $^{95}$Rn displays more pronounced enrichment, indicating that a large volume of isotones with $N = 50$ remains robust. Such a property is due to larger magnetic susceptibility for protons than for neutrons. Following arguments of waiting point approximation, one would expect some slight magnetic effect in the r-process peak with an enhanced portion of small mass number nuclides.

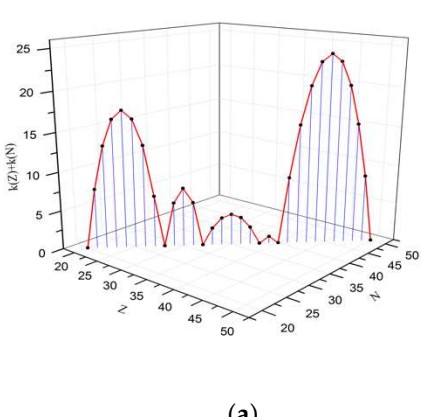

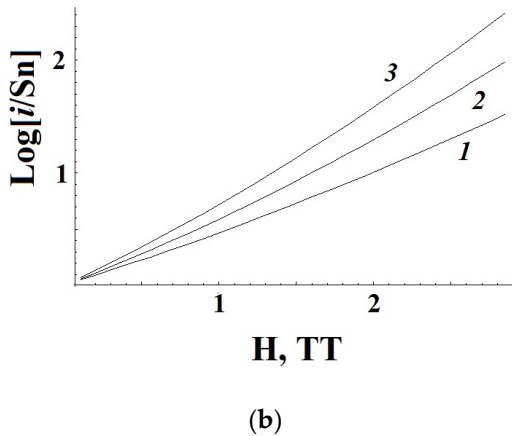

(b)

(a)

**Figure 2.** Nuclear magnetic effects: (**a**) Dependence on numbers of protons and neutrons of the magnetic susceptibility for symmetric nuclei in region *A*=40—100; (**b**) Magnetic field dependence of the yield ratio [*i*/Sn] (see text) for [100]Sn and[96]Cd—1, [92]Pd—2, [95]Rn—3 at kT=0.5 MeV.

## 4. Conclusions

We considered the ultramagnetized atomic nuclei which arise in explosions of type II supernovae, neutron star mergers, during collisions of heavy ions, and in magnetar crusts. It is shown that for a field strength of 0.1—10 TT, the magnetic response of nucleons is determined by the Zeeman effect. Accordingly, dominant linear magnetic susceptibility is represented as the combined reactivity of valent nucleons and binding energy increases for open-shell nuclei. For magic nuclei with closed shells, the binding energy is effectively reduced due to the field-induced additional pressure in a free nucleon gas. As a result, the composition of atomic nuclei formed in an ultramagnetized plasma depends on the field strength. Considerable magnetic modification of nuclear properties is predicted for mass numbers corresponding to large valent shell spins and pronounced magic numbers, i.e., N&Z = 28, 50, 82, 126 . . .

The magnetic structure change for $1f_{7/2}$ shell nuclei (iron group) enhances nucleosynthesis products of smaller mass numbers. In particular, an increase in the volume part of the titanium [44]Ti isotope at a field induction of several TT is in satisfactory agreement with data of direct observations of SN remnants [2,6,8,9]. Such an induction of the magnetic field is consistent with SN explosion energy [2]. These conditions of nucleosynthesis imply a significant increase in a portion of the main titanium isotope,[48]Ti, in the chemical composition of galaxies.

As an example of the synthesis of nuclei with open $1g_{9/2}$ shell and magic number $N = 50$, we see that magnetic effects in the r-process give rise to an enrichment of nuclear components with smaller mass numbers as well. However, a large volume of isotones with $N = 50$ remains robust. Then, a magnetic effect in the r-process peak is expected to result in some enhancement of volume of small mass number nuclides. The magnetic effects considered can also stimulate dynamical deformations in nuclear collisions which are important in subbarrier fusion reactions [18,19] and in the formation, composition, structure, and topology of magnetar inner crusts, see [20].

We notice, finally, that heavy ion collisions giving rise to magnetic fields of ~$10^2$ TT affect quark and gluon dynamics [4], with potential effects on the chiral transition and quarkyonic matter [21] which are important in the experiments being undertaken at the Facility for Antiproton and Ion Research (FAIR) at GSI and the Nuclotron-based Ion Collider fAcility (NICA) at JINR.

**Funding:** This research received no external funding.

**Acknowledgments:** Author (V.N.K.) thanks JINR (Dubna) for the warm hospitality and the financial support.

**Conflicts of Interest:** The authors declare no conflict of interest.

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
