# Peer review of "Properties and Composition of Magnetized Nuclei"

_2571-712X, doi:10.3390/particles3020021_

Round 1

Reviewer 1 Report

This paper concerns the impact of strong magnetic fields on the distribution of iron group nuclei, with relevance to nucleosynthetic yields of these elements. The aim is to reconcile the observations of Titanium isotope enrichment in supernovae and in the chemical composition of galaxies by considering the collective response of nuclei to the magnetic field at the astrophysical site. 

Although the work is presented in a straightforward way, my main concern is that this work is a repetition of the ideas already stated by this author in previous papers, with only slightly different formalism and change of language. Citation [2] (JETP 2015 by same authors) has basically the same conclusion but assumes a spatial profile for the magnetic field, whereas the current work assumes a constant field. Citation [8] (Phys. Lett. B 2018 by same first author) is even more closely related to this paper, with several matching sentences. Another work by the same first author (not cited by this paper for some reason) arrives at the same conclusion about magnetized iron-group nuclei. That work is published in Monthly Notices of the Royal Astronomical Society, Volume 480, Issue 4, p.5380-5383. Through all these works, the conclusions remain the same - that the linear response of valence nucleons acts to enhance (decrease) binding energies for open (closed)-shell nuclei for the range of magnetic field and isotope mass range under consideration. Consequently, there is no purpose to justify publication of essentially the same research result in yet another journal. 

The authors should clearly state what is different in this work from their previous papers, and highlight the novel results presented here, before submitting this paper for publication here or elsewhere.

In addition, the authors have taken a very narrow view of the titanium enhancement issue. There are other plausible explanations for titanium enrichment in supernovae that have not even been referenced, such as 

[1] K. Maeda and K. Nomoto, Astrophys. J. 598, 1163 (2003)

[2] R. Ouyed, D. Leahy, A. Ouyed and P. Jaikumar, PRL 107, 151103 (2011)

Finally, the fact that NuSTAR did not see titanium enrichment in the narrow jet regions associated to CasA casts serious doubt on whether magnetic field effects on nucleosynthesis are the main cause of the enrichment, since magnetic fields corral the jets. 

Given these issues, I cannot recommend publication of this article.

Author Response

Dear Colleague,

Thank you very much for the report regarding the manuscript mentioned above, constrictive criticism and useful suggestions, which are accounted for in the revised version.

Specifically to your comments

Point 1:  “Although the work is presented…”
Response 1: It is right that the methods and technique used in the paper resemble those employed in the refs. mentioned in your report. However, this paper considers also other phenomena, like neutron star mergers and r-process. When it was not stated clearly in previous version, I apologize.  

Point 2: “In addition, the authors have taken… “…
Response 2:Thank you very much for careful reading the manuscript and suggesting references added in the revised version of the manuscript.  

Point 3: “Finally, the fact that NuSTAR did …”
Response 3: Titanium enrichment detected in the narrow jet regions associated to SN1987A supports magnetic field effects on nucleosynthesis are the main cause of the enrichment.  

Main changes in the revised version of the manuscript are given separately.

Thank you very much for your kind attention.        

All the best.

Sincerely,

The author

Reviewer 2 Report

The author analyzes some properties and mass distribution of ultra-magnetized atomic nuclei (which involved in process related with
heavy-ion collisions, magnetar crusts, during Type II supernova explosions and neutron star mergers). In particular, discusses the effect of the magnetic field on various finite nuclei in terms of their binding energies.

Actually, the paper is just a short review of his previous studies (see for example: V.N. Zeeman energy in nucleosynthesis at strong magnetization in supernovae, Kondratyev, MNRAS 480, 5380-5383 (2018)). The author summaries the main points and results without providing more details. In particular, although it is mentioned at the beginning of the the abstract that: "which arise in
heavy-ion collisions, magnetar crusts, during Type II supernova explosions and neutron star
mergers" he does not present details of these applications. Especially, the applications on neutron star merger, is of major interest due to the observational progress the last years. I suggest the author at least to discuss more extensively the issue of the various applications. I suggest also the author to consider the following points:

1)  The draft suffers from many punctuation problems, for example: line 32: remnant.., equation (1) , instead . at the end, line 84: .The method, line 90: .provide e.t.c

2) line 64: the author must to provide the definition of the quantities $\omega_{L}$ and $g_a$ at this point and not later.

3)  line 81: The author must to provide a short comment about the Hund rule (he mentioned the rule without giving any detail) in order o help the reader to understand the comparison.

Author Response

Dear Colleague,

Thank you very much for the report regarding the manuscript mentioned above, constrictive criticism and useful suggestions, which are accounted for in the revised version.

Specifically to your comments

Point 1: “The draft suffers from many …”
Response 1: I did my best to make corrections.  

Point 2:  “line 64: the author must … “…
Response 2: Thank you very much for careful reading the manuscript.  The corrections is done.  

Point 3:  “line 81: The author must …”
Response 3:  I hope it is clear in the revised version.  

Main changes in the revised version of the manuscript are given separately.

Thank you very much for your kind attention.        

All the best.

Sincerely,

The author

Round 2

Reviewer 2 Report

The authors  modified the draft according to my suggestions and comments. I consider that the draft deserves publication in "Particles" in its present form.

Author Response

Response: I agree